# Evaluation of Candidiasis in Upper-Aerodigestive Squamous Cell Carcinoma Patients—A Clinico-Mycological Aspect

**DOI:** 10.3390/ijerph19148510

**Published:** 2022-07-12

**Authors:** Priyanka Debta, Santosh Kumar Swain, Mahesh Chandra Sahu, Abdulwahab A. Abuderman, Khalid J. Alzahrani, Hamsa Jameel Banjer, Ahtesham Ahmad Qureshi, Mohammed Mousa H. Bakri, Gargi S. Sarode, Sangram Patro, Saswati Siddhartha, Shankargouda Patil

**Affiliations:** 1Department of Oral Pathology & Microbiology, Institute of Dental Sciences, Siksha ‘O’ Anusandhan Deemed to Be University, K8, Kalinga Nagar, Bhubaneswar 751003, Odisha, India; debtapriyanka@gmail.com or; 2Department of Otorhinolaryngology, IMS and SUM Hospital, Siksha ‘O’ Anusandhan Deemed to Be University, K8, Kalinga Nagar, Bhubaneswar 751003, Odisha, India; santoshvoltaire@yahoo.co.in; 3ICMR-Regional Medical Research Center, Bhubaneswar 751023, Odisha, India; mchsahu@gmail.com; 4Department of Basic Medical Sciences, College of Medicine, Price Sattam Bin Abdulaziz University, Al-Kharj 16278, Saudi Arabia; a.abuderman@psau.edu.sa; 5Department of Clinical Laboratories Sciences, College of Applied Medical Sciences, Taif University, P.O. Box 11099, Taif 21944, Saudi Arabia; ak.jamaan@tu.edu.sa (K.J.A.); h.banjer@tu.edu.sa (H.J.B.); 6Division of Oral and Maxillofacial Surgery, Department of Oral and Maxillofacial Surgery and Diagnostic Sciences, College of Dentistry, Jazan University, Jazan 45412, Saudi Arabia; drahtesham007@yahoo.co.in (A.A.Q.); mmb644@nyu.edu (M.M.H.B.); 7Department of Oral Pathology and Microbiology, Dr. D. Y. Patil Dental College and Hospital, Dr. D. Y. Patil Vidyapeeth, Pune 411018, Maharashtra, India; gargi14@gmail.com or; 8Department of Oral and Maxillofacial Surgery, Hi-Tech Dental College and Hospital, Bhubaneswar 751007, Odisha, India; drsangrampatro@gmail.com; 9Department of Oral Pathology & Microbiology, Hi-Tech Dental College and Hospital, Bhubaneswar 751007, Odisha, India; saswati20@gmail.com; 10Department of Maxillofacial Surgery and Diagnostic Sciences, Division of Oral Pathology, College of Dentistry, Jazan University, Jazan 45412, Saudi Arabia; 11Centre of Molecular Medicine and Diagnostics (COMManD), Saveetha Dental College & Hospitals, Saveetha Institute of Medical and Technical Sciences, Saveetha University, Chennai 600077, Tamil Nadu, India

**Keywords:** aerodigestive, *Candida*, candidiasis, swab, oral cancer, squamous cell carcinoma

## Abstract

*Candida* is a commensal yeast. It can be infective when the host’s defense mechanism is weakened, as in the case of squamous cell carcinoma patients. We aimed to evaluate the prevalence and clinical mycological manifestation of candidiasis in 150 cancer cases comprised of preoperative and post-operative (with or without radiotherapy) upper aerodigestive squamous cell carcinoma. A total of 150 patients suffering from squamous cell carcinoma of the Upper Aero-Digestive Tract (UADT) were divided into preoperative (*n* = 48), post-operative without radiotherapy (*n* = 29) and post-operative with radiotherapy (*n* = 73). Samples were collected using cotton swabs and cultured. *Candida* species were identified according to color pigmentation on *Candida* Differential Agar (CDA) plate. The clinico-mycological association of patients was evaluated by the chi-square test, and 98 out of 150 patients showed the presence of various *Candida* species. The major species isolated was *Candida albicans* (53%), followed by *Candida tropicalis* (16%). There was a significant statistical difference between patients who showed mycological associations and patients who did not have any such association (*p* = 0.0008). The prevalence of oral candidiasis was found to be 65.33% among total cases of upper aero-digestive squamous cell carcinoma. Chronic erythematous cases of candidiasis were mainly seen in preoperative squamous cell carcinoma cases, whereas the acute erythematous type of candidiasis was mainly seen in post-operative cases who received radiotherapy. The clinicomycological assessment can help to correlate the signs and symptoms with the presence of candidiasis in upper aerodigestive squamous cell carcinoma patients. Meticulous testing and examination can help in the early detection of candidiasis. Future studies are needed to develop advance scientific preventive strategies for high-risk cases.

## 1. Introduction

Fungal infections are a growing worldwide threat that is a major cause of human disease and death [1]. Fungal infections affect more than a million people globally. They are responsible for over 1.5 million deaths annually [2]. *Candida* is a genus of fungi that are normally commensal colonizers of the skin and mucosal surfaces [3]. They are the most common cause of fungal infections worldwide [4]. Long-term global surveillance studies have revealed that five *Candida* species are responsible for the majority of infections detected: *C. albicans* causes most of the candidemia, followed by non-*albicans* strains such as *Candida glabrata*, *Candida tropicalis*, *Candida parapsilosis,* and *Candida krusei* [5,6]. Invasive diseases are generally consequences of a breakdown in the host’s defense mechanism [7,8]. Infections caused by *Candida* species have risen dramatically worldwide, driven by an increase in the number of immunocompromised patients [9,10].

Certain cancers and their treatment, including surgical and radiotherapy, can create immunocompromised situations, predisposing the patient to opportunistic infections [11]. Elevated incidence of *Candida* infections is evident in carcinoma patients [12]. *C. albicans*, considered to be the most serious cause of candidiasis, is diploid, polymorphic yeast producing three morphologic forms: yeast cells, pseudo hyphae, and true hyphae. Radiation therapy to the head and neck region can cause xerostomia, thereby increasing the incidence of oral candidiasis [13].

Head and neck cancer, including Oral Squamous Cell Carcinoma (OSCC), is the sixth leading cancer worldwide, with an estimated 300,400 cases and 145,400 OSCC-related death occurring on an average in a year [14]. Its incidence varies geographically and is related to a combination of environmental and genetic factors [15,16]. A history of tobacco use, nitrosamine, and alcohol intake are also potential risk factors [17,18]. Candidiasis is very frequently manifested in esophageal carcinoma patients (27%). Several studies hypothesize the causal role of fungi on oral carcinogenesis [19,20,21]. The catalytic activity of *Candida* is thought to promote the production of carcinogenic nitrosamines [22,23]. Components of the yeast cell wall may modulate proinflammatory cytokine release, enhancing the cancer microenvironment [24].

Though the pathogenic mechanisms and carcinogenic ability of *Candida* remain unclear, it appears that a significant relationship exists between oral cancer and the degree of *Candida* colonization observed. Several studies have reported the prevalence of various species of *Candida* in the saliva of squamous cell carcinoma patients. Mäkinen et al. found several *Candida* species in 74% of oral cancer patients’ saliva samples. Perera et al. observed a dysbiotic microbiome dominated by *Candida* sp. in association with oral squamous cell carcinoma patients. It was found that *Candida* was present in greater frequency in patients with oral cancer compared with non-cancer controls [25]. This is in contrast with the results of Sanketh et al., who reported that only 10 OSCC cases out of 100 were positive for *Candida* [26].

There is a relative paucity of studies focusing on the clinico-mycological assessment of *Candida* sp. in oral cancer cases. This warrants further investigation into the prevalence, mycobiome profile, and mycological manifestation of *Candida* in squamous cell carcinoma patients. A systematic understanding of the prevalence and pathogenesis of *Candida* species in aerodigestive squamous cell carcinoma patients remains unclear. Our study examines the various species of *Candida* isolated from upper aerodigestive squamous cell carcinoma patients and evaluates the relationship between the various oral signs and symptoms with respect to mycological manifestations of the patients.

## 2. Materials and Methods

### 2.1. Sample Collections

The present study was carried out in an Odisha-based population in India.

Ethical permission was taken to examine samples for identification of various candida species in the research laboratory of the Institute of Medical Sciences, SOA, BBSR, Odisha, India (Ref. No. DMR/IMS.SH/SOA/180254).

Cancer patients suffering from Squamous cell carcinoma of the upper aerodigestive tract (UADT), such as oral and oropharyngeal squamous cell carcinoma, were included in this study. Written informed consents were taken from patients after explaining the nature of the study and that their oral and oropharyngeal swab sample was taken for evaluation of *Candida* fungal infection.

Swab samples were collected from 150 cancer patients who reported to the Institute of Medical Sciences (IMS) and SUM Hospital, Odisha, India, pre-operatively for surgery and post-operatively for follow-up. Pre-operative and post-operative cases, with or without radiotherapy, showed signs and symptoms of candidiasis. Various signs and symptoms such as dry mouth complaints of cases were confirmed by sialometric determination by establishing a resting or unstimulated whole saliva flow of <0.1–0.2 mL/min and a stimulated whole saliva flow of <0.4–0.7 mL/min.

Poor oral hygiene was observed by evaluating teeth, gums, and tongue. Halitosis was a common finding due to poor oral hygiene, whereas pain and dysphagia were symptoms noted in cancer cases. The presence of white patches and redness of oral mucosa were observed and noted during the evaluation of the oral cavity. Clinical signs and symptoms were examined, followed by sample collection, followed by the investigation procedure of laboratory to see the presence or absence of candidiasis and to see various candida species in the sample which showed the presence of candidiasis.

The lesions present were mostly in the form of white patches, reddish patches, or ulcerative in nature. They were found in various sites of the oral cavity such as buccal mucosa, retromolar area (Figure A1), tongue, and oropharyngeal area such as tonsillar area (Figure A2), and pharyngeal wall. Swabs were collected from pre- and post-operative squamous cell carcinoma patients’ mouth. Post-operative patients were further categorized as with or without radiotherapy after surgery.

Inclusion criteria:(1)Primary cases of upper-aerodigestive squamous cell carcinoma patients (pre- and post-operative cases).(2)Post-operative cases with both categories, such as with or without radiotherapy.(3)Patients who had not taken antifungal therapy before the swab sample collection.(4)Patients showing signs and symptoms of candidiasis, such as dysphagia, poor oral hygiene, dry mouth, altered taste sensation, halitosis, pain, whitish patch, and reddish patches.

Exclusion criteria:(1)Patients who had systemic diseases were excluded from the study.(2)Patients under drug therapy of long-term antibiotics and corticosteroids were excluded from the study.

### 2.2. Sample Examination

The samples were collected with the help of two cotton swabs by moving them against the lesions. One of the swabs was evaluated to check the presence of yeast cells by cytosmear. The specimen was inoculated with sabouraud dextrose agar with the help of the other swab and incubated for 2 days. Among 150 swab samples, the samples isolate that showed budding yeast colonies with cytosmear were cultured on sabourad Dextrose Agar (SDA) plates and incubated at 37 °C for 48 h. A single colony from the SDA plate was cultured on *Candida* Differential Agar (CDA) plate. After 24–48 h of incubation, color pigmentation of *Candida* species was observed on the CDA plate.

### 2.3. Statistical Analysis

Data management and analysis were performed using SPSS v.18.0 (Build 1.0.0.1275, Creator—IBM, Chicago, IL, USA). The demographic details of the patients were tabulated in the form of tables and graphs. For the statistical analysis of the clinico-mycological association of candidiasis, a chi-square test was used. The results were considered significant if *p <* 0.05.

## 3. Results

The study included a total of 150 cancer patients, out of which 100 were oral carcinoma patients, and 50 belonged to oropharyngeal carcinoma. The demographic distribution of patients with age, sex, and site is presented in Table 1.

Among all patients, those in the 61–70 age groups presented with more squamous cell carcinoma cases than the other age groups. Male patients were 60% of the total cases, while females comprised 40%. As per site-wise distribution of oral and oropharyngeal squamous cell carcinoma cases, OSCC of the retromolar area (Figure 1) showed more involvement, and 60% of cases had a chewing habit history (Table 2).

The patients were evaluated for the clinical signs and symptoms of candidiasis. Dysphagia was mostly seen in cases followed by dry mouth (Table 3). By performing chi-square test; *p* was found to be 0.0008 between the mycologically positive and mycologically negative patients with various symptoms. Table 3a shows the difference in such symptoms between the mycologically positive and negative patients was found to be statistically significant (*p =* 0.0008, *p <* 0.05).

Of the total 150 patients, 98 samples (65.33%) showed the presence of candidiasis with various candida species (Table 4). Of the total patients shown in Table 5, 48 cases were pre-operative, 29 cases were post-operative without radiotherapy, and 73 cases were post-operative with radiotherapy.

Table 4 describes the species-wise distribution of *Candida* isolates, suggesting that 53.06% were *C. albicans* and the rest were non-albican species such as *Candida tropicalis, Candida glabrata, Candida krusei, Candida parapsilosis,* and *Candida dubliniensis. Candida* species were identified according to color pigmentation. *C. albicans*—light green, *C. tropicalis*—blue to purple, *C. glabrata*—cream to pinkish-white, *C. krusei*—fuzzy purple, *C. parapsilosis*—white to cream, *C. dubliniensis*—pale green. The prevalence of oral candidiasis was found to be 62.5% in pretreatment cases and 68.96% in post-operative cases without radiotherapy, and 65.75% in post-operative patients after receiving radiotherapy (Table 5).

Various types of candidiasis were observed in the study group—Acute pseudomembranous, Acute erythematous and Candida-associated lesion (Denture stomatitis), Candida-associated lesion (Angular cheilitis), Chronic pseudomembranou, and Chronic erythematous candidiasis (Figure 2). Among all 150 upper-aerodigestive squamous cell carcinoma cases with clinical manifestation, 98 cases showed candidiasis.

## 4. Discussion

Patients who undergo therapy for head and neck cancers are at an increased risk for opportunistic infections due to their weakened immune systems [27]. A debilitating side effect of surgical and radiotherapy is the immune dysfunction that follows, leaving cancer patients vulnerable [28]. Several species of *Candida* are commensals, colonizing the oral mucosa. However, compromised host defenses can turn these fungi into infectious agents [29]. Candidiasis and mucositis can cause immense discomfort and alter a patient’s quality of life. In this study, we sought to determine the prevalence of *Candida* species in the swab of patients with oral and oropharyngeal squamous cell carcinoma.

We discovered that a majority of cancer patients (65.33%) in our study were positive for various *Candida* species. Other studies also has been observed that candida species colonize the aerodigestive tract of 70% of the total cancer patients [30,31]. T cells are responsible for suppressing the virulence of *Candida* in the mucosal epithelium, so when there is a breakdown of these mucosal barriers in cancer patients, the opportunistic candidiasis sets in [32]. High morbidity and mortality are normally associated with invasive candidiasis [33]. Ramirez Amador et al. observed that 38% of such patients show oropharyngeal candidiasis [34]. Studies show that from 7 to 52% of cancer patients undergoing chemo and radiotherapy do develop candidiasis [35]. These findings are in line with the results of our study. The optimal radiation dose depends on the size and location of the primary tumors and the neck lymph nodes. In general, primary tumors and gross lymphadenopathy require a total of 70 Gy or more, with a daily fraction of 2 Gy. Radiation to low-risk neck nodal regions requires a total of 50 Gy or more. For post-operative radiotherapy, higher doses of radiation (from 60 to 66 Gy) are generally required for the microscopic disease to decrease the risk of locoregional failure [36].

We evaluated the association of clinical manifestations with the mycological manifestations of candidiasis. We found that carcinoma of the retromolar region was the most prevalent, followed by that of the buccal mucosa region. Poor oral hygiene and dysphagia were the most common symptoms. We found a statistically significant (*p =* 0.0008) difference between the mycological negative and positive cases with respect to various clinical manifestations of candidiasis. Finding the association of clinical signs and symptoms with mycological manifestations can help clinicians identify the cause of the symptoms and help them to decide on the appropriate treatment pattern [37].

This study showed *Candida albicans* to be the most common species found among cancer patients, followed by that *Candida tropicalis* which was similar to that of the studies carried out by Schelenz and Safdar et al. [38,39]. The most common non-*Candida* species among cancer patients was *Candida glabrata,* as reported in some studies [40,41,42,43,44]. This is also in agreement with the findings of Abidullah et al., who reported the presence of *C. glabrata* in patients with oral squamous cell carcinoma [40]. This finding broadly supports the work of Hulimane et al. and Roy et al., who observed *C. glabrata* and *C. albicans* existing in a polyfungal population in OSCC patients [41,42]. These non-*C. albicans* (NCA) are implicated in treatment resistance and deteriorating oral health [42]. Our results are in accord with previous studies that have examined the presence of *Candida* sp. in carcinoma patients [44].

In the present study, the most common non-*albicans* variant we found in patients was *Candida tropicalis*, followed by *Candida glabrata, Candida krusei, Candida parapsilosis,* and *Candida dubliniensis.* This is significant as *Candia* can release aldehydes that may induce mitochondrial damage and increase the release of reactive oxygen species. It can also affect cytosine methylation and DNA repair enzymes [45,46]. Our results are in combination with Sankari et al.’s findings which showed a predominance of *C. krusei* and *C. tropicalis* among OSCC patients [47]. Castillo et al. reported that isolates of *Candida* from OSCC patients exhibited higher attributes of virulence, with non-*albicans* species showing higher biofilm formation [48].

The *Candida* species distribution may show geographical variation. *C. glabrata* was found in 18% of cases in North America [49,50]. In contrast, *C. tropicalis* and *C. parapsilosis* were commonly detected in cases from four Latin American countries [51]. Surveillance programs stretching over twenty years have identified a decrease in the isolation of *C. albicans* and an increase in the isolation of *C. glabrata* and *C. parapsilosis* over time [52]. These isolates are emerging as resistant to antifungal drugs and increasing the burden of invasive fungal infections. Their higher prevalence could be an outcome of an increased population of immunocompromised patients.

There was a significant difference between the signs and symptoms of patients having mycological manifestations and those who did not show any such manifestations. Hence there is a need for routine periodic surveillance of fungal infections in cancer patients undergoing surgical treatment/radiotherapy, showing a high prevalence of *Candida* as observed in our study. Poor oral hygiene, immunocompromised state of carcinoma patients, and radiation-induced hyposalivation are thought to be major predisposing factors that increase the number of candidal species in the oral and oropharyngeal area, eventually leading to candidiasis [53,54]. In the present observational study, we found little difference in the prevalence of candidiasis in post-operative patient groups. Prevalence of candidiasis was 68.9% in post-operative cases without radiotherapy, whereas in post-operative cases with radiotherapy, the prevalence was 66.75%. Onset, types, severity of candidiasis can be varied due to individual immune status, post-operatively psychological stress, and oral hygiene care of patients.

Determining the mycobiome profiles and their correlations with squamous cell carcinoma may help develop screening and surveillance tools for high-risk patients [55]. Impeccable oral hygiene measures supplemented with antiseptic mouth rinses may help prevent candidiasis in immunocompromised individuals [56]. Patients with an increased prevalence of yeasts may require its suppressive treatment before the commencement of anti-cancer therapy. Clinico-mycological evaluation helps in deciding an appropriate mode of treatment. This study provides insights into the signs and symptoms of upper aerodigestive carcinoma patients having candida fungal infections and their association.

Microorganisms such as HPV (human papilloma virus) proved to play an important role in carcinogenesis [57], whereas fungi such as candida mainly emerged as one of the opportunistic microorganisms in cancer patients. Upper aerodigestive candidiasis is also seen in other immunocompromised state, such as in patients with deep neck space infection [58]. Therefore, careful observation and quick identification with an early start of antifungal treatment prevents further progression of candidiasis and improves patient’s condition.

Further research on the association of Candida with oral squamous cell carcinoma will shed light on the molecular mechanisms of disease development. Future studies can develop a fuller picture of inter and intra-kingdom interactions of the species in the mycobiome with neoplasms. Further large-scale studies on this topic with larger sample sizes need to be undertaken at various cancer care hospitals in India.

## 5. Conclusions

Post-surgical treatment cases have shown more association with oral candidiasis in cancer patients. Patients with oral squamous cell carcinoma fall into a high-risk category with respect to opportunistic candidiasis infections. Meticulous testing and examination can improve the detection of oral and oropharyngeal candidiasis. Clinicians should remain vigilant for signs and symptoms such as white patch development and dry mouth so that early lab diagnosis of particular species of candida facilitates early and effective treatment. Future studies can translate scientific advances to develop preventative strategies for high-risk cases. 

## Figures and Tables

**Figure 1 ijerph-19-08510-f001:**
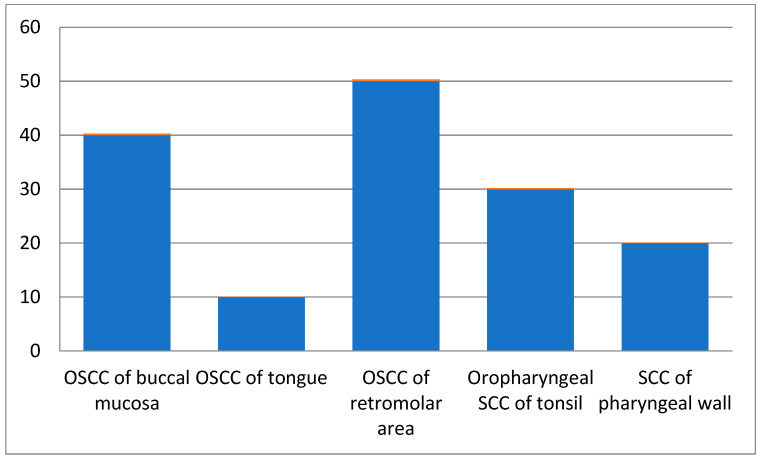
Site-wise distribution of carcinoma cases.

**Figure 2 ijerph-19-08510-f002:**
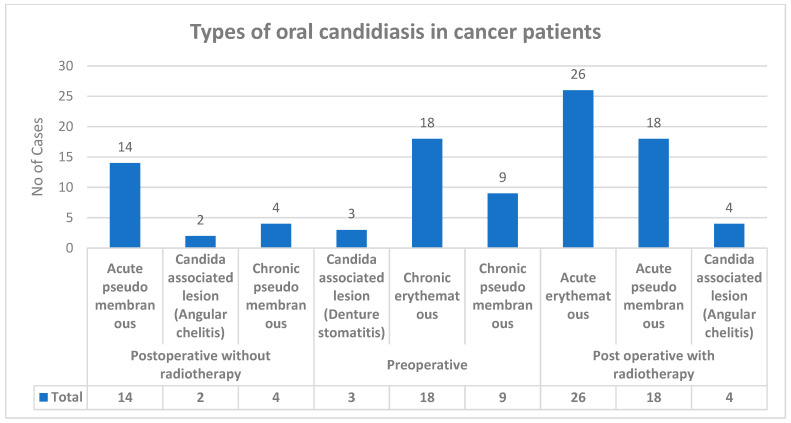
Types of oral candidiasis in carcinoma cases.

**Table 1 ijerph-19-08510-t001:** Demographic distribution of cancer cases.

Demographic Detail	Number of Cases	Percentage
**Age**		
0–20	5	3.3%
21–40	11	7.3%
41–60	54	36%
61–70	80	53.4%
**Sex-wise distribution**		
Male	90	60%
Female	60	40%
**Site-wise distribution of carcinoma cases**		
OSCC of buccal mucosa	40	27%
OSCC of tongue	10	7%
OSCC of retromolar area	50	33%
SCC of tonsil	30	20%
SCC of pharyngeal wall	20	13%

**Table 2 ijerph-19-08510-t002:** Habits of patients.

Habits	Number of Patients	Percentage
Tobacco chewing	90	60%
Tobacco smoking	42	28%
Mixed habit	18	12%
Total	150	100%

**Table 3 ijerph-19-08510-t003:** (**a**) Clinico-mycological evaluation. (**b**) Shows data summary with SD, SE and *p*-value.

(**a**)						
Clinical Manifestations	Total Patient Showing Lesion	Mycologically Positive Patients	Percentage	Mycologically Negative Patients	Percentage	*p*-Value
Dysphagia	67	26	38.8%	41	61.2%	0.0008
Poororal hygiene	72	11	15.2%	61	84.7%	
Dry mouth	58	32	55.1%	26	44.8%	
Altered taste sensation	20	9	45%	11	55%	
Halitosis	32	3	9%	29	90.6%	
Pain	28	3	10.7%	25	89.2%	
Presence of white patch/plaque	20	10	50%	10	50%	
Redness in the mucosa	37	4	10.8%	33	89.1%	
(**b**)						
**Data Summary**						** *p-* ** **Value**
**Groups**		**Symptoms**	**Mean**	**Std Dev**	**Std Error**	0.0008
Group1(mycologically positive)		8	29.25	19.6305	6.9404	
Group2(mycologically negative)		8	70.625	19.6973	6.964	

**Table 4 ijerph-19-08510-t004:** Species-wise distribution of *Candida*.

Variant	Percentage
*Candida albicans*	52 (53.06%)
*Candida tropicalis*	16 (16.32%)
*Candida glabrata*	12(12.24%)
*Candida krusei*	8 (8.16%)
*Candida parapsilosis*	4 (4.01%)
*Candida dubliniensis*	6 (6.12%)
Total	98 (100%)

**Table 5 ijerph-19-08510-t005:** Frequency of oral candidiasis in cancer cases.

Cancer Cases	Number of Cases	Candidiasis Positive Cases	Percentage of Candidiasis
Pre-operative	48	30	62.5%
Post-operative without radiotherapy	29	20	68.96%
Post-operative with radiotherapy	73	48	65.75%
Total cases	150	98	65.33%

## Data Availability

Not applicable.

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
