# Peer review of "Evaluation of Candidiasis in Upper-Aerodigestive Squamous Cell Carcinoma Patients—A Clinico-Mycological Aspect"

_ijerph, 2022, doi:10.3390/ijerph19148510_

Round 1

Reviewer 1 Report

The article is interesting with satisfactory numbers in the file. The statistical methods are sound and statistics is adequate. However, some weak points must be addressed.

1.       The structure of the article is strange and difficult to understand, please restructure.

2.       The Material and methods are after discussion. Is this journal requirement?

3.       I would appreciate some standard deviations or errors mentioned in the text.

4.       The numbers are not easy to follow, there are some of them not easy to understand for example: row 130-131. The prevalence of oral candidiasis was found to be 62.5% in pre-treatment cases and 68.96% in postoperative cases without radiotherapy and 65.75% in patients after receiving radiotherapy.

It is surprising, that radiotherapy patient have less candidiasis than non-radiotherapy patients, because it is usually opposite. It should be explained.

5.       What is standard dose of radiotherapy on tumour and to lymphatics? Add this information.

6.       The occurrence of Candida in upper air and swallowing pathways in imunompromised patients is known not only in oncological cases. I would recommend to add. This information to discussion. For example, in deep neck space infections: Mejzlik J, Celakovsky P, Tucek L, Kotulek M, Vrbacky A, Matousek P, Stanikova L, Hoskova T, Pazs A, Mittu P, Chrobok V. Univariate and multivariate models for the prediction of life-threatening complications in 586 cases of deep neck space infections: retrospective multi-institutional study. J Laryngol Otol. 2017 Sep;131(9):779-784. doi: 10.1017/S0022215117001153. Epub 2017 Jun 5. PMID: 28578716.

Author Response

Comments and Suggestions for Authors

Reply to reviewers…below point wise....

Reviewer 1

The article is interesting with satisfactory numbers in the file. The statistical methods are sound and statistics is adequate. However, some weak points must be addressed.

  1. The structure of the article is strange and difficult to understand, please restructure.
  • Done
  1. The Material and methods are after discussion. Is this journal requirement?
  • Yes sir rearranged as per journal format.

  1. I would appreciate some standard deviations or errors mentioned in the text.

  • Yes sir, After Table 3, Standard deviation and error included.

  1. The numbers are not easy to follow, there are some of them not easy to understand for example: row 130-131. The prevalence of oral candidiasis was found to be 62.5% in pre-treatment cases and 68.96% in postoperative cases without radiotherapy and 65.75% in patients after receiving radiotherapy.-
  • Sir pre-treatment cases group were carcinoma cases before their surgery.

  • Dear sir, it is observed in our study that among 29 postoperative cases 20 cases showed candidiasis whereas 48 cases shown positive among 73 postoperative cases who received radiotherapy.
  • Mam/sir we have included the actual results as obtained in our observational study. In percentage form, post operative cases without radiotherapy, prevalenceof candidiasis was 68.9% whereas in post operative cases with radiotherapy the prevalence was 66.75%.

  1. It is surprising, that radiotherapy patient has less candidiasis than non-radiotherapy patients, because it is usually opposite. It should be explained –

Yes sir, it has been added as….In presentobservational study, we found little difference in prevalence of candidiasis in post operative patient’s groups. Prevalence of candidiasis was 68.9% in post operative cases without radiotherapywhereas in post operative cases with radiotherapy the prevalence was 66.75%.Onset, types, severity of candidiasis can be varied due to individual immune status, post-operatively psychological stress and oral hygiene care of patient.

  1. What is standard dose of radiotherapy on tumour and to lymphatics? Add this information.
  • Yes sir , it is included now in manuscript as..The optimal radiation dose depends on the size and location of the primary tumors and the neck lymph nodes. In general, primary tumors and gross lymphadenopathy require a total of 70 Gy or more, with a daily fraction of 2 Gy. Radiation to low-risk neck nodal regions requires a total of 50 Gy or more. For post operative radiotherapy, higher doses of radiation (60 to 66 Gy) are generally required for microscopic disease to decrease the risk of locoregional failure. [36]

  1. The occurrence of Candida in upper air and swallowing pathways in imunompromised patients is known not only in oncological cases. I would recommend to add. This information to discussion. For example, in deep neck space infections: Mejzlik J, Celakovsky P, Tucek L, Kotulek M, Vrbacky A, Matousek P, Stanikova L, Hoskova T, Pazs A, Mittu P, Chrobok V. Univariate and multivariate models for the prediction of life-threatening complications in 586 cases of deep neck space infections: retrospective multi-institutional study. J Laryngol Otol. 2017 Sep;131(9):779-784. doi: 10.1017/S0022215117001153. Epub 2017 Jun 5. PMID: 28578716.

Yes sir, added as ]Upper aerodigestive candidiasis is also seen in other immunocompromised state like in patients with deep neck space infection.So careful observation and quick identification with early start of treatment prevents further progression and improve patient’s condition [58].

Reviewer 2

  • The submitted manuscript is interesting and aimed to evaluate the role of Candida spp. in the oncogenesis of the upper aerodigestive tract.
  • Thank you so much sir.
  • The text requires an English language and grammar revision.
  • Grammar revision done. Edited mistakes.
  • The taxonomy must be revised and rules applied in the whole text (e.g., italics for the mname of the microorganisms, use of capital letter for the genus name only, but not for the species).
  • Yes sir, Taxonomy corrected.
  • In Introduction, please consider that HPV is also involved in the carcinogenesis of the upper aero-digestive tract. Read and discuss the following: doi: 10.1016/j.disamonth.2018.09.007; doi: 10.1002/cre2.435;doi: 10.3390/jof7060476; doi: 10.3390/v13040559.
  • Ye sir included in discussion end part as….. Micro-organism like HPV (human papilloma virus) proved to play important role in carcinogenesis [57]. Whereas fungi like candida is mainly emerges as one of the opportunistic microorganisms in cancer patient. whereas fungi candida albicans is mainly emerges as one of the opportunistic microorganisms in cancer patient.

  • How the authors diferentiated the Candida species? It was by the pigmentation of culture growth only?  Please better describe this, and read and discuss the following: doi: 1186/1476-0711-5-1; doi:10.1002/yea.891; doi: 10.4103/jomfp.JOMFP_157_18;     doi: 10.1186/1471-2334-12-230; doi: 10.1159/000493426.

  • Sir as per;  doi: 10.1186/1471-2334-12-230. They suggested molecular method for differentiating candida albicans and candida dubliniensis as these species share almost same phenotypic features. But in routine practice if the lab following biosafety and fumigation and sterilization care properly so careful examination of phenotypic features is sufficient to identify the various species of candida as it is gold standard basic procedure and cost effective.
  • Sir till now many studies conducted with work of phenotypic identification of candida species in various immunocompromised patients as it is not possible to do molecular work due to finance issue and their work published in reputed journals.Sir, this present study is part of PhD work, in future we will do molecular work also.

General comment-

  • Please revise the manuscript according to the referees' comments and upload
    the revised file within 10 days.

    Please use the version of your manuscript found at the above link for your

    (I) Please check that all references are relevant to the contents of the
    manuscript.
    (II) Any revisions to the manuscript should be marked up using the “Track
    Changes” function if you are using MS Word/LaTeX, such that any changes can
    be easily viewed by the editors and reviewers.
    (III) Please provide a cover letter to explain, point by point, the details
    of the revisions to the manuscript and your responses to the referees’
    comments.
    (IV) If you found it impossible to address certain comments in the review
    reports, please include an explanation in your rebuttal.
    (V) The revised version will be sent to the editors and reviewers.

  • Reply to reviewer prepared within 10 days.

  • Cover letter and reply to reviewer is made.

  • Thank you.

Reviewer 2 Report

The submitted manuscript is interesting and aimed to evaluate the role of Candida spp. in the oncogenesis of the upper aerodigestive tract.

The text requires an English language and grammar revision.

The taxonomy must be revised and rules applied in the whole text (e.g., italics for the mname of the microorganisms, use of capital letter for the genus name only, but not for the species).

In Introduction, please consider that HPV is also involved in the carcinogenesis of the upper aero-digestive tract. Read and discuss the following: doi: 10.1016/j.disamonth.2018.09.007; doi: 10.1002/cre2.435; doi: 10.3390/jof7060476; doi: 10.3390/v13040559.

How the authors diferentiated the Candida species? It was by the pigmentation of culture growth only?  Please better describe this, and read and discuss the following: doi: 10.1186/1476-0711-5-1; doi: 10.1002/yea.891; doi: 10.4103/jomfp.JOMFP_157_18;     doi: 10.1186/1471-2334-12-230; doi: 10.1159/000493426.

Author Response

(The authors gave the same response as above.)

Round 2

Reviewer 1 Report

This manuscript was significantly improved

Reviewer 2 Report

The authors addressed all previous suggestions and replied with clear and acceptable explanations for not-revised parts.

The manuscript now appears suitable for the journal.